# Electrical Properties of Two-Dimensional Materials Used in Gas Sensors

**DOI:** 10.3390/s19061295

**Published:** 2019-03-14

**Authors:** Rafael Vargas-Bernal

**Affiliations:** Departamento de Ingeniería en Materiales, Instituto Tecnológico Superior de Irapuato (ITESI), Carretera Irapuato-Silao Km. 12.5, Irapuato, Guanajuato 36821, Mexico; ravargas@itesi.edu.mx; Tel.: +52-462-606-7900 (ext. 123)

**Keywords:** mathematical modeling, gas sensors, two-dimensional materials, graphene, transition metal dichalcogenides, field effect transistors, chemiresistors

## Abstract

In the search for gas sensing materials, two-dimensional materials offer the possibility of designing sensors capable of tuning the electronic band structure by controlling their thickness, quantity of dopants, alloying between different materials, vertical stacking, and the presence of gases. Through materials engineering it is feasible to study the electrical properties of two-dimensional materials which are directly related to their crystalline structure, first Brillouin zone, and dispersion energy, the latter estimated through the tight-binding model. A review of the electrical properties directly related to the crystalline structure of these materials is made in this article for the two-dimensional materials used in the design of gas sensors. It was found that most 2D sensing materials have a hexagonal crystalline structure, although some materials have monoclinic, orthorhombic and triclinic structures. Through the simulation of the mathematical models of the dispersion energy, two-dimensional and three-dimensional electronic band structures were predicted for graphene, hexagonal boron nitride (*h*-BN) and silicene, which must be known before designing a gas sensor.

## 1. Introduction

Since the isolation of graphene in 2004, numerous two-dimensional materials have been discovered, isolated, synthesized [1,2,3,4,5,6,7,8,9,10,11,12,13,14,15], and/or developed using computational tools. Researchers around the world have shifted their research focus from the syntheses processes to the modification of two-dimensional materials to optimize their electronic properties in the design of emerging electronic devices such as solar cells, supercapacitors, field effect transistors (FETs), and gas sensors [1,2,3,4,5,6,7,8,9,10,11,12,13,14,15,16,17,18,19,20,21,22,23,24,25,26,27,28,29,30,31,32]. Among the electronic properties that can be controlled is the structure of energy bands which can be achieved through: (1) alloying between two-dimensional materials to form two-dimensional ternary materials, (2) vertical stacking of two-dimensional materials that can produce 2D heterostructures [12,13,14], known as van der Waals solids, and (3) controlling the thickness of two-dimensional materials through their number of layers. In the high-tech manufacture of electronic devices, the use of two-dimensional materials such as graphene, transition metal dichalcogenides, black phosphorus, hexagonal boron nitride (*h*-BN), silicene, germanene, stanene, arsenene, aluminene, antimonene, bismuthine, molybdenum disulfide (MoS_2_), molybdenum diselenide (MoSe_2_), MXenes, etc. [1,2,3,4,5,6,7,8,9,10,11,12,13,14,15] These materials have attracted the attention of gas sensor designers due to their large surface-to-volume ratios and extremely sensitive surfaces [16,17,18,19,20,21,22,23,24,25,26,27,28,29,30,31,32,33,34,35,36,37,38,39,40,41,42,43,44,45,46,47,48,49,50,51,52,53,54,55,56,57,58,59,60,61,62,63,64,65,66,67,68,69,70,71,72,73,74,75,76,77,78,79,80,81,82,83,84,85,86,87,88,89,90,91,92,93,94,95,96,97,98,99,100,101,102,103,104,105,106,107]. Graphene has been investigated more exhaustively as the sensitive material for the development of chemical sensors by exploiting the sensing structure based on a field effect transistor, which has shown excellent gas sensing capacity [1,2,7,8,10,16,18,22,23,26,29,32,33,34,35,36,100,101,102,103,104,105,106,107]. However, the inherent disadvantage of graphene is its zero bandgap, which reduces its sensitivity and selectivity to a wide range of analytes. To optimize these capabilities of the gas sensors, vertically aligned two-dimensional structures [77,106], surface chemical functionalization [20,92], as well as two-dimensional heterostructures [12,13,14] and nanocomposites [26,27,72,107,108] have been developed. Unfortunately, these techniques do not favor long-term stability in the gas sensors, so by adding substitutional impurities [33,83] in the crystalline lattice of the two-dimensional material, the electrical conductivity of the sensitive material is modulated, and reliable and reproducible sensors can be implemented.

Recently, two-dimensional materials have been used for gas sensing because their structure in thin layers with atomic thickness offers large surface-to-volume ratios and large absorption capacity of gas molecules thanks to their intense surface activity [16,17,18,19,20,21,22,23,24,25,26,27,28,29,30,31,32]. Additionally, 2D materials have the potential to exist in more than one crystalline structure which increases the possibility that materials may present either a semiconducting phase (2H) (such as GaSe, In_2_Se_3_, MoS_2_, MoSe_2_, MoTe_2_, SnS_2_, WS_2_) or a semi-metallic phase (1T’) (such as TaS_2_) which makes the band gap can be tunable, they have a high sensitivity to a wide variety of chemicals, and their thickness modifies the physical and chemical properties of the material [18]. The enormous challenge of the use of two-dimensional materials is the efficient integration of these materials with three-dimensional systems, since this can limit the performance of electronic devices and their use in circuits or systems.

This article has been divided as follows: Section 2 presents an overview of the two-dimensional materials used for the design of gas sensors from a point-of-view associated with materials engineering. A special emphasis is given to the electrical properties, the crystalline structure and lattice parameters, as well as to establish an exhaustive list of gases that have been detected by two-dimensional sensing material. Further, the advantages of using 2D materials in gas sensing are analyzed. A study of the electronic band structure of two-dimensional materials for the design of gas sensors is carried out in Section 3, making use of the knowledge of the first Brillouin zone and the dispersion energy of these materials. The electronic band structure of graphene, hexagonal boron nitride and silicene are obtained through the tight-binding model, which demonstrates that the use of mathematical modeling and its simulation must be applied to all sensing materials to optimize the design of gas sensors. In Section 4, the importance of the study of band engineering is discussed as a strategy that must be matured for two-dimensional sensing materials. Finally, the conclusions of the article are given in Section 5.

## 2. Two-Dimensional Materials Used in Gas Sensors

2D materials are referred as materials with a single layer or a few layers of material. A layer can be defined as a material with single thickness covering a surface. They can be classified either as allotropes of unique chemical elements (graphene, silicene, germanene, etc.) or with various chemical or compound elements (*h*-BN, dichalcogenides, MXenes, etc.); the latter consisting of two or more covalently bonding elements. The layers are stacked together via van der Waals interactions and can be exfoliated into thin 2D layers. The combinations of layers of different two-dimensional materials are generally called van der Waals heterostructures. These materials have been applied mainly in photovoltaic devices, semiconductor devices, electrodes for supercapacitors, and in water purification. A more extensive description of the various applications of two-dimensional materials can be found in [6,8] in previous publications by the same author. Main applications of the two-dimensional materials in electronic industry are enumerated in Figure 1. A summary of the electrical properties, crystal structure and lattice parameters of the main two-dimensional materials used in electronic applications is presented in Table 1 for materials with a single chemical element, and in Table 2 for materials based on chemical compounds.

### 2.1. Two-Dimensional Materials for Gas Sensing

A gas sensor is a device which detects the presence and/or quantifies the concentration of one or various gases within a specific volume by means of chemical and/or physical interaction of the gas or gases with a sensing material for determining any change in a physical property before and after of this action [17]. The mechanism of gas sensing is basically based on the transfer of electric charge between the chemical species to be detected and the sensing material which acts as an absorber or donor, in this case the two-dimensional material. This charge transfer will cause changes in the electrical properties of the sensing material. For example, when the sensing material is of the *n*-type (electron donor), its electrical resistance increases in the presence of oxidizing gases or decreases when reducing gases are detected. The opposite is presented for *p*-type materials or electron acceptors. Commonly, the structure used in gas sensing devices is based on field effect transistors or FETs, in which the sensing material is placed between the source and drain electrodes supplied by a constant voltage. Gas detection is performed by monitoring changes in the value of the electric current between the gate and the source due to the exposure of a gas to be detected. The main performance parameters of a gas sensor are related to its response time, selectivity, sensitivity, stability and recovery time [17]. The great advantage of two-dimensional materials is that they can be easily doped with catalytic materials that can improve the absorption of gases and thereby improve the selectivity of the gas sensor. Main properties of the two-dimensional materials used in gas sensors are shown in Figure 2.

Although gas sensors were developed since the last century, the possibility of having gas sensors for excessively low concentration levels, with shorter response times, with higher sensitivity and better selectivity, good stability, high repeatability, as well as operating at room temperature; nowadays, these capabilities continue being a challenge to be solved in the technological world of today. Lately, researchers around the world are applying two-dimensional nanometric materials as an alternative to previous, present and future developments to overcome these challenges. A list of two-dimensional materials that have been used in gas sensing is summarized in Table 3.

A composite material is a material made from two or more constituent materials with significantly different physical or chemical properties that, when combined, produce a material with different characteristics from the individual components. Different composite materials based on two-dimensional materials have been used to design gas sensors [26,27,72,102,103,104,105,106,107]. Metals, polymers and ceramics have been added as a second phase to these two-dimensional materials. Layered combinations of different 2D materials are generally called van der Waals heterostructures or hybrid two-dimensional materials. Hybrid two-dimensional materials are advanced multifunctional materials that have outstanding physical and chemical properties and are extensively considered for applications such as energy storage, energy conversion, energy harvesting technologies, and sensors [101,108]. A list of composite and/or hybrid materials using two-dimensional materials [101,102,103,104,105,106,107] that have been used in gas sensing is summarized in Table 4.

### 2.2. Advantages of Two-Dimensional Materials for Gas Sensing

The two-dimensional materials used as gas sensing materials are particularly interesting because [16,17,18,19,20,21,22,23,24,25,26,27,28,29,30,31,32]: (1) their different active sites of interest such as defects, vacancies and edge sites, which allow for selective molecular absorption, (2) their surface-to-volume ratios, and (3) their high yield preparations. Chemical doping by means of noble metals (e.g., silver (Ag), gold (Au), palladium (Pd), platinum (Pt)) and/or metals with low working function (e.g., scandium (Sc), yttrium (Y)) can be used to tune the electronic properties of two-dimensional materials [109], see Figure 3.

Two-dimensional materials, other than graphene, have a bandgap that can be tunable by thickness, which makes them ideal channel materials for sensing using field effect transistors [24,29,75]. Capabilities of sensitive material such as high carrier mobility and high on/off ratio are also important for the implementation of gas sensors. The number of layers of the two-dimensional material is directly associated with the stability and it is inversely proportional to the response time. The main gas sensing mechanism is the charge transfer reaction between the gaseous species and the two-dimensional material.

Other advantages that two-dimensional materials offer in gas sensing are higher thermal stability and wider operation temperature range [18]. 2D nanostructures can provide some additional advantages including more active sites, facile surface functionalization, good compatibility with device integration, possibility to be assembled into 3D architectures, etc. which are critically important for developing high performance gas sensors [25]. Further, hybrid structures based on two-dimensional materials can be designed to optimize the selectivity and sensitivity of the gas sensors.

Gas sensors based on heterostructures overcome the inherent disadvantages of gas sensors based on simple materials such as poor selectivity and insensitivity to the low concentration of the gas to be detected [110]. Moreover, 2D materials can be easily fabricated as chemiresistive field effect transistors (FETs) that consume less power and offer good safety [21]. Thanks to these last capabilities, the two-dimensional materials are distinguished from conventional metal oxides in that they offer an excessively sensitive platform and ideal for portable applications due to the reduced power consumption.

## 3. Electronic Band Structure for Two-Dimensional Materials

The electrical properties of a solid are determined basically through its electronic band structure, which establishes the range of energy states that electrons may or may not have within their crystalline structure. The allowed electronic states are represented by the valence band (states within the atom) and the conduction band (stable states with freedom of movement). Between the valence band and the conducting band, there is a set of electronic states that are not allowed or prohibited for electrons, since these are intermediate and unstable. The electronic band theory predicts these bands by solving the Schrödinger equation for a periodic crystal structure making use of the quantum mechanical wave functions or Bloch functions of the electrons. Particularly, solid state devices based on semiconductor materials such as field effect transistors [24,29,75], solar cells, gas sensors, [16,17,18,19,20,21,22,23,24,25,26,27,28,29,30,31,32] etc., owe their electronic behavior to the exploitation of these electronic bands. A summary describing the design process of a gas sensor through the electrical properties of the sensing material is illustrated in Figure 4.

The solution of the Schrödinger equation, in addition to describing the changes in the time of the behavior of electrons, also relates the state vector of its quantum system (***ψ***), its position vector (***r***), and the periodicity function of the crystal (*u*). This can be mathematically expressed as [111,112]:(1)$$\psi_{nk}\left(r\right)\;=\;e^{ik\cdot r}u_{nk}\left(r\right)$$
where ***k*** is called the wavevector and ***n*** represents a band index. Therefore, its multiple solutions *E_n_*(***k***) represent the *n* energy bands evaluated for each wavevector ***k*** that establish the energy dispersion relationships of the electrons in the crystal lattice.

According to the crystalline structure of a material, the wavevector ***k*** takes values within what is called the Brillouin zone, which establishes the states within the electronic band structure. The Brillouin zone has a symmetry that can be identified by the points and lines that relate the different crystallographic directions in the material, which are denoted as Γ or [000], Δ or [100], Λ or [111], and Σ or [110]. Theoretically, obtaining a graph of the behavior of the energy *E* against the components of the wavevector ***k***: *k_x_*, *k_y_*, *k_z_*, implies a four-dimensional space that connects the points of symmetry. But in a practical way, two-dimensional graphs of the structure of bands that are isosurfaces of constant energy in the wavevector space are feasible for all states with an energy value.

The energy bands of electronic materials can be classified according to the types of wavevectors in two different types: (1) direct bandgap: where the space of prohibited energy is defined between the lowest-energy state of the band conduction and the highest-energy state of the valence band when both have the same wavevector, and (2) indirect bandgap: where the space of prohibited energy is defined between the lowest-energy state of the conduction band and the highest-energy state of the valence band when both states have different wavevectors.

The periodic crystalline structures can be defined by a Bravais lattice formed by a set of vectors ***R*** = *n*_1_***a***_1_ + *n*_2_***a***_2_ + *n*_3_***a***_3_ where *n_i_* are any integers and ***a**_i_* are primitive vectors forming the lattice. Moreover, for each Bravais lattice a reciprocal lattice can be identified by means of the reciprocal vectors ***k*** = *m*_1_***b***_1_ + *m*_2_***b***_2_ + *m*_3_***b***_3_ where *m_i_* are any integers and *b_i_* are reciprocal vectors forming the reciprocal lattice [111,112,113]. In this way, mathematical expressions relating the Bravais lattice and the reciprocal lattice are given below [111,113]:(2)$$\begin{array}{c} b_{1}\;=\;2\pi \frac{a_{2}\times a_{3}}{a_{1}\cdot \left(a_{2}\times a_{3}\right)}\\ b_{2}\;=\;2\pi \frac{a_{3}\times a_{1}}{a_{2}\cdot \left(a_{3}\times a_{1}\right)}\\ b_{3}\;=\;2\pi \frac{a_{1}\times a_{2}}{a_{3}\cdot \left(a_{1}\times a_{2}\right)}. \end{array}$$

The physical dimensions of the unit cells forming a crystalline lattice can be defined by three lattice constants or parameters (*a*, *b*, *c*) and three interaxial angles (*α*, *β*, and *γ*), as shown in Figure 5.

There are seven different crystalline systems. They differ by having different relations between unit cell axes and angles. The relations between the length of the unit cell axes and the angles between them are given in Table 5. The symmetry and stability of the crystalline lattices may be increasing from triclinic, via monoclinic, orthorhombic, hexagonal, tetragonal or rhombohedral to the cubic system [111,112,113].

### 3.1. First Brillouin Zone of Two-Dimensional Materials

In the reciprocal space, the first Brillouin zone is the only primitive cell that can be defined. However, different areas of Brillouin can be represented to completely model the reciprocal lattice of a crystalline material. The first Brillouin zone represents crystallographically the values of Bloch waves directly associated with the electronic states in the crystal lattice.

In the first Brillouin zone as well as in the Bravais lattice, different points of symmetry can be identified, which are called critical points, which define the unique vector space of each crystal structure. The critical points for the crystalline lattices found in the two-dimensional materials with hexagonal lattice are summarized in Table 6.

A point group can be defined as a group of geometric symmetries (isometries) that keep at least one point fixed. The three-dimensional point groups are commonly used in chemistry to describe the crystalline symmetries expressed by means of rotation axes. Rotation axes are denoted by Hermann-Mauguin notation that uses a number *n*: 1, 2, 3, 4, 5, 6, 7, 8 … (angle of rotation *φ* = 360°/*n*). Rotoinversion axes are defined using a number *n* with a macron, $\bar{n}$:$\bar{1}$, $\bar{2}$, $\bar{3}$, $\bar{4}$, $\bar{5}$, $\bar{6}$, $\bar{7}$, $\bar{8}$ …. $\bar{2}\;$which is equivalent to a mirror plane, and generally expressed as *m*. In addition, there is a crystallographic restriction theorem which states that rotational symmetries are normally restricted to 2-fold, 3-fold, 4-fold, and 6-fold.

The first Brillouin zone of the crystalline lattices found in 2D materials [114] is shown in Table 7.

#### 3.1.1. First Brillouin Zone for materials with Hexagonal Crystalline Lattices

Graphene is a two-dimensional material consisting of carbon atoms in a hexagonal lattice [1,2,7,8,10,16,18,22,23,26,29,32,33,34,35,36,90,101,102,103,106]. Other two-dimensional materials with hexagonal crystalline lattice are germanene [37,38,39,40,41,42,43], silicene [1,2,22,29,45,46,47], stanene [2,48,49,50,51,104], aluminene [64], bismuthene, antimonene [29,65,66,67,68], hexagonal boron nitride (*h*-BN) [83,84,85], gallium sulfide (GaS), gallium selenide (GaSe), hafnium disulfide (HfS_2_) [82], hafnium diselenide (HfSe_2_) [82], indium selenide (In_2_Se_3_), molybdenum disulfide (MoS_2_) [32,75,76,77,78,79,101], molybdenum ditelluride (MoTe_2_) [82], molybdenum diselenide (MoSe_2_) [80,81], molybdenum sulfide selenide (MoSSe), molybdenum tungsten diselenide (MoWSe_2_), antimony telluride (Sb_2_Te_3_), tin disulfide (SnS_2_) [94,95], tin selenide (SnSe_2_) [96], tantalum disulfide (TaS_2_), tungsten disulfide (WS_2_) [32,71,72,73], tungsten selenide (WSe_2_) [74], and zirconium diselenide (ZrSe_2_). The first Brillouin zone of a hexagonal lattice is summarized in Table 8.

In addition, $\bar{\Gamma A}\;=\;\frac{\pi }{c}$, $\bar{\Gamma K}\;=\;\frac{4\pi }{3a}$, $\bar{\Gamma M}\;=\;\frac{2\pi }{\sqrt{3}a}$, and $\bar{MK}\;=\;\frac{2\pi }{3a}$. 

#### 3.1.2. First Brillouin Zone for Materials with Orthorhombic Crystalline Lattices

Two-dimensional materials with orthorhombic crystalline lattice are borophene (striped), black phosphorus (BP), germanium sulfide (GeS), and dibismuth trisulphide (Bi_2_S_3_). The first Brillouin zone of an orthorhombic lattice is summarized in Table 9.

In addition, $\bar{\Gamma Y}\;=\;\frac{\pi }{a}$, $\bar{\Gamma Z}\;=\;\bar{YT}\;=\;\bar{SR}\;=\;\frac{\pi }{c}$, and $\bar{\Gamma T}\;=\;\frac{\pi }{ac}\sqrt{a^{2}\;+\;c^{2}}$.

#### 3.1.3. First Brillouin Zone for Materials with Triclinic Crystalline Lattices

Two-dimensional materials with triclinic crystalline lattices are rhenium disulphide (ReS_2_) and rhenium diselenide (ReSe_2_). The first Brillouin zone of a triclinic lattice is summarized in Table 10, and whose values only can be determined knowing all values of the lattice parameters and interaxial angles.

#### 3.1.4. First Brillouin Zone for Materials with Monoclinic Crystalline Lattices

Two-dimensional materials with monoclinic crystalline lattices are diarsenic tritelluride (As_2_Te_3_) and zirconium triselenide (ZrSe_3_). The first Brillouin zone of a P monoclinic lattice is summarized in Table 11.

In addition, $\bar{\Gamma Y}\;=\;\bar{ZD}\;=\;\frac{2\pi }{b\sin\gamma }$, $\bar{XA}\;=\;\frac{2\pi }{c}$, and $\bar{\Gamma X}\;=\;\bar{AZ}\;=\;\frac{2\pi }{a}\sqrt{1\;+{\;\tan}^{2}\gamma }$.

### 3.2. Tight-Binding Model for Two-Dimensional Materials

To predict the electronic energy bands, the tight binding model can be used. This model assumes that the electrons in the crystal behave as an assembly of constituent atoms. Further, under the use of this model, the solution to the time-independent single electron Schrödinger equation can be approximated by a linear combination of atomic orbitals. This model works well on material with limited overlap between atomic and potential orbitals on neighboring atoms.

#### 3.2.1. Band Structure of Graphene

Graphene is an allotrope of carbon made from a simple layer of atoms with a hexagonal crystal structure. It is a semimetal with small overlap between the valence and the conduction bands (zero bandgap material). The tight-binding model (or TB model) is a mathematical model used to determine the electronic band structure by means of a set of wave functions of isolated atoms located at atomic sites of the crystal structure. Tight-binding models are applied to a wide variety of solids. Tight binding dispersion relation for graphene is given as (http://lampx.tugraz.at/~hadley/ss1/bands/tbtable/tbtable.html) [115,116]:(3)$$E\;=\;\varepsilon \pm t\sqrt{1\;+\;4\cos\left(\frac{\sqrt{3}k_{x}a}{2}\right)\cos\left(\frac{k_{y}a}{2}\right)\;+\;4\cos^{2}\left(\frac{k_{y}a}{2}\right)}$$
where *E* is the energy of the system depending on the ***k***-vectors, *ε* is the ionization energy of the atom of the unit cell (here carbon), *t* is the overlap integral, *a* is the lattice constant, and *k_x_* and *k_y_* are the *k*-vectors in *x* or *y* direction. Tight binding dispersion relation for graphene is shown in Figure 4. For *ε* = 0 eV, *t* = 2.8 eV and *a* = 1.421 Å, *E* was estimated as illustrated in Figure 6. Bandgaps open at the M-points between the first and the second bands. From here on, the level of the valence band is shown in blue and the level of the conduction band in green. No bandgaps open at the ***k***-points and the ***k***’-points as shown in Figure 7.

#### 3.2.2. Band Structure of Hexagonal Boron Nitride (*h*-BN)

Hexagonal boron nitride (*h*-BN) has a layered structure like graphene. Within each layer, boron and nitrogen atoms are bound by strong covalent bonds, with boron atoms being up and down the nitrogen atoms. Tight binding dispersion relation for two-dimensional boron nitride is given as (http://lampx.tugraz.at/~hadley/ss1/bands/tbtable/tbtable.html) [117]:(4)$$E\;=\;\frac{\varepsilon_{1}\;+\;\varepsilon_{2}}{2}\pm \sqrt{\frac{{\left(\varepsilon_{1}-\varepsilon_{2}\right)}^{2}}{2}\;+\;4t^{2}\left(cos\left(\frac{\sqrt{3}k_{x}a}{2}\right)cos\left(\frac{k_{y}a}{2}\right)\;+\;4cos^{2}\left(\frac{k_{y}a}{2}\right)\;+\;\frac{1}{4}\right)}$$
where *E* is the energy of the system depending on the *k*-vectors, ε_1_ and ε_2_ are the ionization energies of the two kinds of atoms of the unit cell (here boron and nitrogen), *t* is the overlap integral, *a* is the lattice constant, and *k_x_* and *k_y_* are the *k*-vectors in *x* or *y* direction. Tight binding dispersion relation for hexagonal boron nitride is shown in Figure 8. For *h*-BN, where *ε*_1_ = 2 eV, *ε*_2_ = 1 eV, *t* = 2 eV and *a* = 1.47 Å, *E* was estimated as illustrated in Figure 9.

#### 3.2.3. Band Structure of Silicene

Silicene is the two-dimensional allotrope of silicon with a hexagonal crystalline structure like that of graphene [1,2,22,29,45,46,47]. Tight binding dispersion relation for two-dimensional silicene is given as [118]:(5)$$E\left(k\right)\;=\;a_{10}^{55}\pm \left[a_{61}^{55}\left|k\right|\;+\;a_{11}^{55}{\left|k\right|}^{2}\;+\;a_{62}^{55}k_{x}\left(3k_{y}^{2}-k_{x}^{2}\right)/\left|k\right|\right]$$
where $\left|k\right|\;=\;\sqrt{k_{x}^{2}\;+\;k_{y}^{2}}$, *k_x_* and *k_y_* are the *k*-vectors in *x* or *y* direction, $a_{10}^{55}$, $a_{61}^{55}$, $a_{11}^{55}$, and $a_{62}^{55}$ are coefficients obtained by the tight binding approximation of the band structure of silicene. Tight binding energy dispersion relation for silicene is shown in Figure 10, for *k_y_* = *k_x_*/$\sqrt{3}$ and $\frac{\Delta z}{\Delta \mathrm{Si}}\;=\;0.59$. For silicene in three-dimensional way, *E* was estimated as illustrated in Figure 11.

## 4. Why Study Electrical Properties of the 2D Materials for Gas Sensors?

The band gap of any material determines the energy required by an electron to jump from the top level of the valence band (where it is not mobile enough to conduct freely) to just reach the ground level of conduction band (where, as its name suggests, the electron is able to conduct freely). Its electrical properties (or maybe a singular property) must be different in the presence of gas compared to what they were in the absence of gas. Thus, the band gap and those technical variables that modify its value are directly associated with the electrical conductivity of the sensing material.

The nanometric gas sensors based on conductance and fabricated with two-dimensional materials are attractive thanks to their higher sensitivity/selectivity and relatively low cost [18]. Semiconducting 2D materials exhibit better sensitivity than insulating/metallic 2D materials. The study of the dispersion energy of the first Brillouin zone allows to tune the band gap of two-dimensional material in the search for semiconducting materials with the maximum performance in the gas sensing. Through the mathematical models of dispersion energy, it is possible to study the effect of the thickness and the inclusion of dopants on the band gap of the two-dimensional material [33,34,35,36,37,38,39,40,41,42,43,44,45,46,47,48,49,50,51,52,53,54,55,56,57,58,59,60,61,62,63,64,65,66,67,68,69,70,71,72,73,74,75,76,77,78,79,80,81,82,83].

Band-gap engineering is the area of engineering that studies the process of controlling or altering the band gap of a material [3]. This is typically done to semiconductor materials by controlling their composition or constructing layered materials with alternating compositions. Controlling the band gap of a semiconductor material allows the creation of desirable electrical properties. By varying the composition of the two-dimensional material, it is possible to alter the band gap of the resulting material, because the bonding of the atoms of the original sensing material with atoms of different nature of the doping material produces forbidden bands with different values for sensing oxidizing gases or reducing gases.

The exposure of the sensing material to different gases, different concentrations, and different conditions leads to a different sensor response, since the electrical conductivity presented by the gas sensor is unique. The use of mathematical models to predict the electronic band structure allows to study the electrical conductivity that will be presented by a gas sensor, where parameters such as material size, chemical compositions, doping, etc. will be incorporated, reduces the design time of the sensor. In Figure 12, a sketch of the methodology proposed in this paper to design gas sensors based on two-dimensional materials is shown.

In addition, the band gap of semiconducting materials can be modified by inducing strain because quantum confinement is modified [50]. A range of upper tunable bandgaps is possible because two-dimensional materials have higher yield strength. Bending by strain can induce an increase in electrical conductance [79]. The strain can also induce a change in the transport properties and a variation in the band gap. The strain can be converted into a work function that modifies the band gap of the two-dimensional material. The work function can be added directly to the energy dispersion model as a summative term. The narrower the ribbon of the two-dimensional material, the greater the opening of the forbidden gap of the material since it is considered a quasi-one-dimensional system in which a semiconducting behavior occurs.

The operating temperature plays a very important role since it defines the temperature at which the maximum sensitivity of the sensor is reached and determines the detection limit of the gas sensor [17]. In portable applications, the reduction of operating temperature of the gas sensors has a direct effect on the reduction of the energy required to bring the sensing material to the operating temperature. The improvement of the adsorption of gases on the two-dimensional sensing material lies in the vacancies of high-energy defects found on the surface of the material [95]. Additionally, the adsorption energies are relatively low, which allows them to trap gas molecules more easily [119]. Another important factor for two-dimensional materials can sense gases at low temperatures are that the mobility of electrons in them is excessively high in this temperature range, which makes them very sensitive.

### 4.1. Correlation between Band Gap and/or Electronic Band Structure and Electrical Conductivity

The two-dimensional materials have attracted the attention of various groups of scientists around the world to be used in gas sensing, due to the ratio of surface area to large volume, good thermal and chemical stability, and sensitivity of electrical properties to changes in the environment that surrounds them [18]. In the case of graphene, its derivatives such as graphene oxide (GO) and reduced graphene oxide (rGO) have been used mainly because they have a non-zero band gap. The enormous advantage of two-dimensional materials is that they allow the development of highly selective and excessively sensitive sensors by adjusting their surface chemistry without modifying their optical and electrical properties. Thanks to the fact that these materials have a high electrical conductivity and low Johnson electrical noise, the modification of the concentration of carriers induced by their exposure to gases produces significant changes in electrical conductivity or resistivity.

The presence of Stone-Wales defects or dopants in two-dimensional materials increases the adsorption of gases due to the change in the band gap of them. This change is because they reduce the amplitude of the bandgap and thereby increase the electrical conductivity of the sensing material [33,120]. The electrical conductivity of two-dimensional materials, either occurring naturally due to a semiconductor behavior or induced by modifications such as doping, defects or chemical functionalization, is directly related to the band gap space. In doped semiconductors, the dopants increase the charge carrier’s concentration by adding electrons to the conduction band or at increasing the quantity of holes in the valence band. Therefore, the addition of dopants, defects or chemical functionalization modifies the electronic band structure of the sensing material, either increasing or decreasing the size of the band gap. The smaller the band gap space, the greater the electrical conductivity of the sensing materials. The band gap of two-dimensional materials is illustrated in Figure 13 [121].

Other two-dimensional materials of group IV such as germanene, silicene and stanene have a higher chemical reactivity than graphene to be used as gas sensors due to their buckling structure [49]. This structure causes the stanene to show linear band dispersion at the Fermi energy which allows it to tune its band gap by means of spin-orbit coupling (SOC). In addition, the stanene behaves as a spin-quantum Hall insulator that when functionalized can increase its band gap, thereby presents an electrical dissipationless conductivity at room temperature, making it a highly sensitive material for gas sensing applications. These properties induce p-, n- and co-doping to improve the interaction between stanene and gas molecules [122]. Even, the co-doping changes the electronic properties at the Fermi level, which is highly desired in two-dimensional materials that have semi-metallic behavior.

In the case of two-dimensional materials based on transition metal dichalcogenides (TMD), such as MoS_2_, MoSe_2_, MoTe_2_, WS_2_, WSe_2_, etc., they are potential options to be used as sensing materials thanks to the fact that they present semiconducting phase (2H) and semi-metallic phase (1T’), which allow producing materials with tunable bandgaps, according to the needs of the gas sensor [4,6,7,15,20,32]. Additionally, its physical and chemical properties are dependent on the thickness or number of layers of two-dimensional material used in the design [123,124]. By reducing the thickness of the two-dimensional material, it is even possible to modify the size of the forbidden band or even modify the type of forbidden band from indirect to direct or vice versa according to the type of two-dimensional material used as sensing material. In very thin two-dimensional materials there are no quantum confinement effects and the electronic structure is dominated by surface states near the Fermi level.

### 4.2. Correlation between Gas Sensing Characteristics and Electronic Band Structure

Since the selectivity of a gas sensor is related to its ability to respond to a gas in the presence of others, then the design of a gas sensor implies that the gas to be detected must be adsorbed by the sensing material, exclusively, and this is achieved, if the necessary catalysts are added to this material [32]. The catalytic materials must be based on metallic nanoparticles. These materials modify the band gap and, in this way, only those gases with ionization energy like the band can be adsorbed.

The adsorption energy (*E_a_*) of the gases is calculated using the following mathematical expression [49]:(6)$$E_{a}\;=\;E_{sensing\;material\;+\;gas}-\left(E_{sensing\;material}\;+\;E_{gas}\right)$$
where $E_{sensing\;material\;+\;gas}$ implies the total energy of the system formed by the sensing material and the sensed gas. On the other hand, $E_{sensing\;material}$ and $E_{gas}$ represent the punctual energy of the two-dimensional material based system and gas molecules, respectively.

Since the sensitivity of a gas sensor expresses the change in the output signal per unit gas concentration, it is necessary to design a material capable of detecting levels up to below parts per billion of gas present in the sensing material [17]. Two-dimensional materials have a high sensitivity due to their high surface-to-volume ratio and their semiconductor properties. Both the dimensions of the sensing material and the semiconductor properties of the material can be tuned; in this way, the sensitivity of a gas sensor based on two-dimensional materials can be designed. The band gap of the sensing material plays a major role in adjusting the sensitivity of the gas sensor. Besides, the doped two-dimensional materials show higher selectivity and sensitivity toward gas molecules compared to pure two-dimensional materials [49]. The sensitivity and selectivity of a semiconductor two-dimensional material based gas sensor depend on any change in its electronic properties.

Another of the important performance parameters of gas sensors is reversibility, which implies that the sensor must be capable of being used during a cyclic operation without qualitatively or quantitatively modifying its response to the target gas [17]. This is committed to selectivity, since when the latter is high; the reversibility is low due to the high bond energies involved [93]. Therefore, complete reversibility is achieved when weak interactions between the gas and the sensing material are present [125]. Then, a two-dimensional material with a bandgap without catalysts may have better reversibility. This can be appreciated when there are no intermediate bands or levels between the valence band and the conduction band.

Gas sensors require a response time to react to the presence of the target gas whose concentration changes from zero to a certain concentration value [17]. This response time is directly related to the type of band gap that the two-dimensional material has, since being a direct band gap would be expected to have a shorter response time than a material with an indirect band gap. Although the path between the valence band and the conduction band that the electron should travel energetically was straight, the magnitude of the band gap should be short to reduce the response time. A small response time is directly associated with a high sensitivity of the gas sensor, which is linked to a reduced band gap [122].

In any physicochemical process such as gas sensing, stability represents the ability of the sensor to keep reproducible its performance, in a specific period of time, before various physicochemical variables [17]. Since gas sensors are susceptible to being modified by various variables such as temperature, then the bandgap of the two-dimensional material must remain stable before any event presented. Some gas molecules in contact with the sensing material tend to dissociate and chemically absorb into it. Other gases tend to be physically absorbed stably on the sensing material, which leads to different interaction strengths. These latter molecules induce different modifications in the forbidden band of the material, which can be observed in the energy band diagram [52].

Key performance scores for gas sensors based on two-dimensional materials are illustrated in Figure 14. A more exhaustive study on optimizing the performance parameters of a gas sensor such as sensitivity, selectivity, stability, response time and operating temperature is found in [17].

## 5. Conclusions

Despite the great advances in the application of two-dimensional materials in gas sensing, the study of the fundamental properties of these materials as well as the development of mathematical models to predict the electronic band structure is still in its infancy. There is a wide variety of 2D materials that can be used to sense both oxidizing gases and reducing gases. This article studies the electronic band structure of 2D materials to optimize gas sensing through their first Brillouin zone and dispersion energy. It was shown that most two-dimensional materials used in gas sensors have a hexagonal crystalline structure, and that the tight-binding model can optimize the electronic band structure through mathematical modeling and its simulation. It is considered important to resume the study of two-dimensional materials using the perspective of materials engineering, and to implement a wide variety of gas sensors using the different topologies of materials that have been proposed until now and propose new design options.

## Figures and Tables

**Figure 1 sensors-19-01295-f001:**
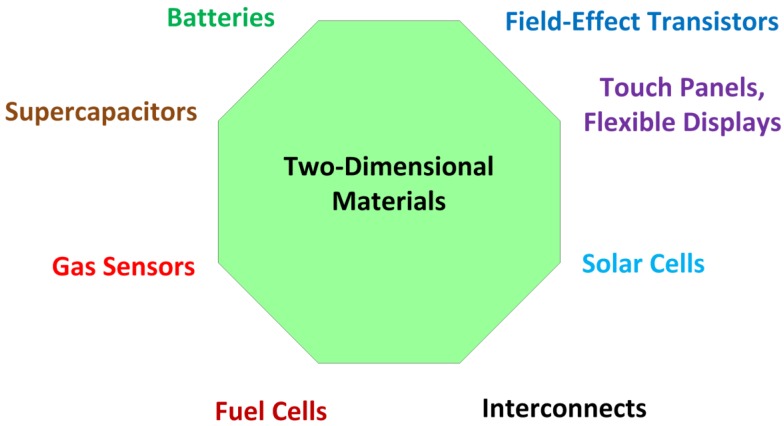
Main applications of the 2D materials in electronic industry.

**Figure 2 sensors-19-01295-f002:**
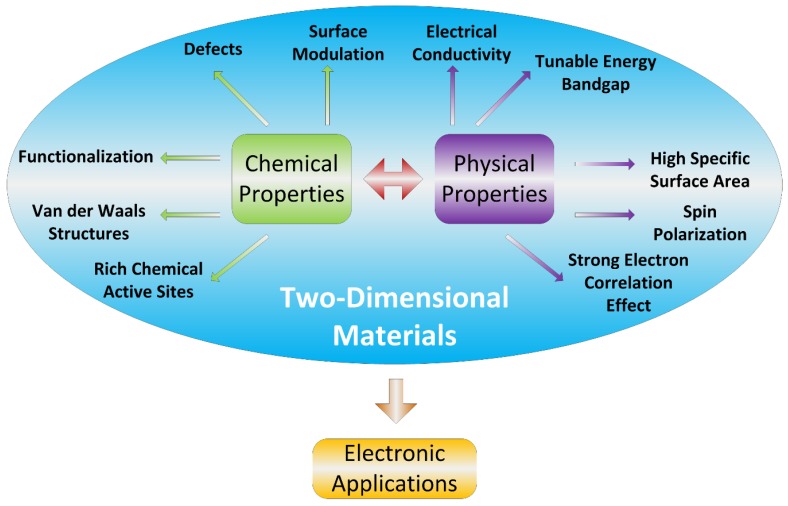
Main properties of the two-dimensional materials in gas sensing.

**Figure 3 sensors-19-01295-f003:**
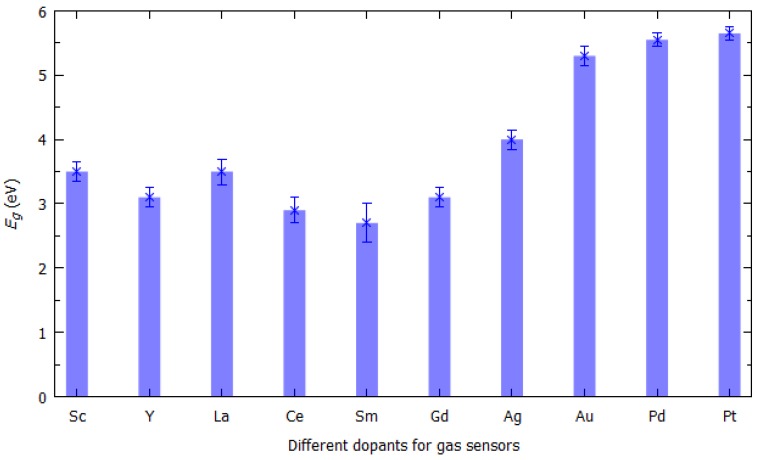
Experimentally measured values of work function for different metals.

**Figure 4 sensors-19-01295-f004:**
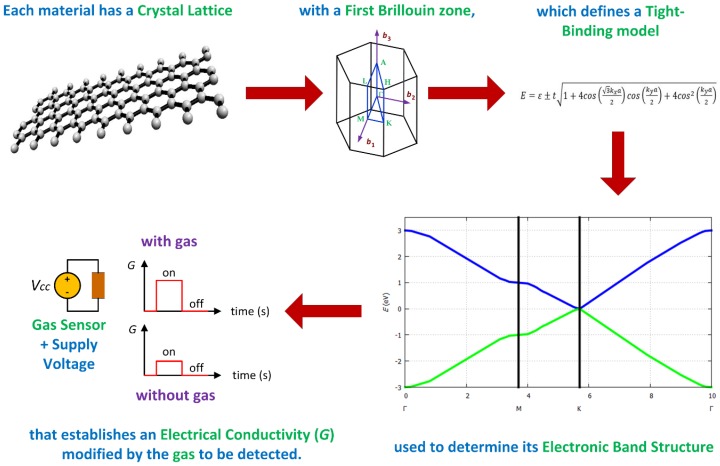
Design process of a gas sensor through the electrical properties.

**Figure 5 sensors-19-01295-f005:**
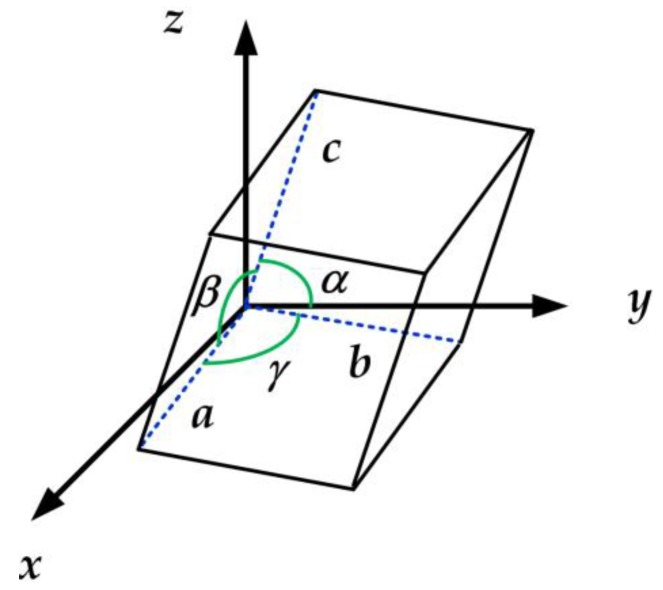
Parameters in a crystalline lattice.

**Figure 6 sensors-19-01295-f006:**
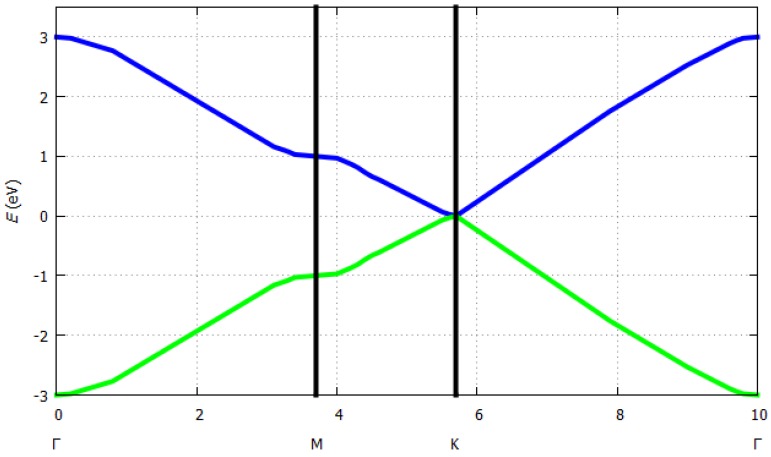
Band structure of a single graphene layer (http://lampx.tugraz.at/~hadley/ss1/bands/tbtable/dispgraphene.html?).

**Figure 7 sensors-19-01295-f007:**
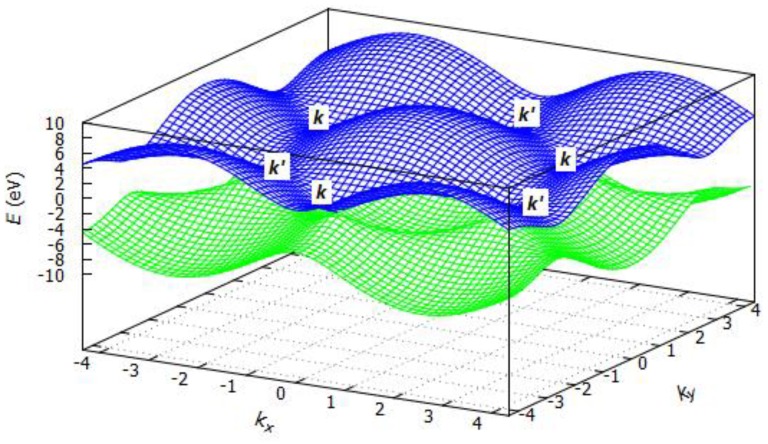
Graphene energy band structure.

**Figure 8 sensors-19-01295-f008:**
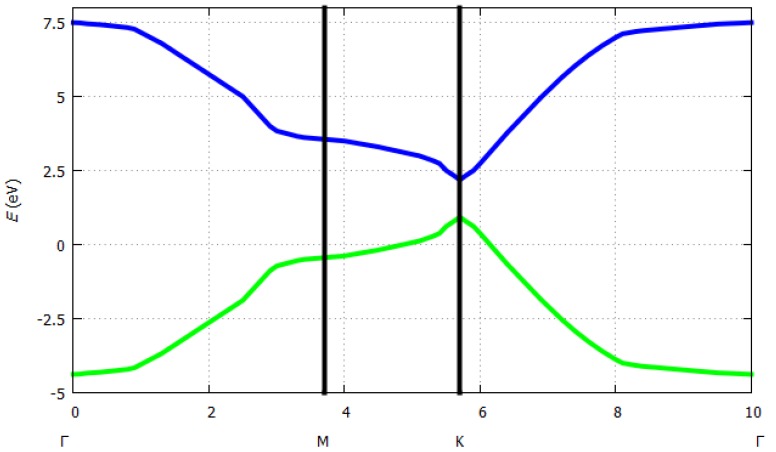
Tight binding dispersion relation for two-dimensional boron nitride (http://lampx.tugraz.at/~hadley/ss1/bands/tbtable/dispbn.html?).

**Figure 9 sensors-19-01295-f009:**
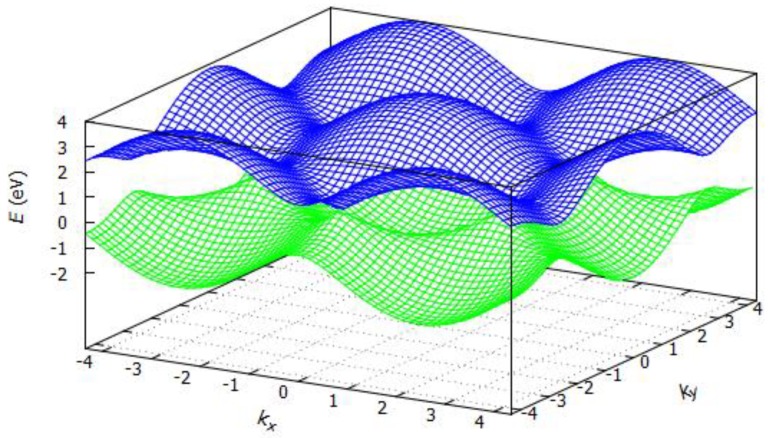
Hexagonal boron nitride energy band structure.

**Figure 10 sensors-19-01295-f010:**
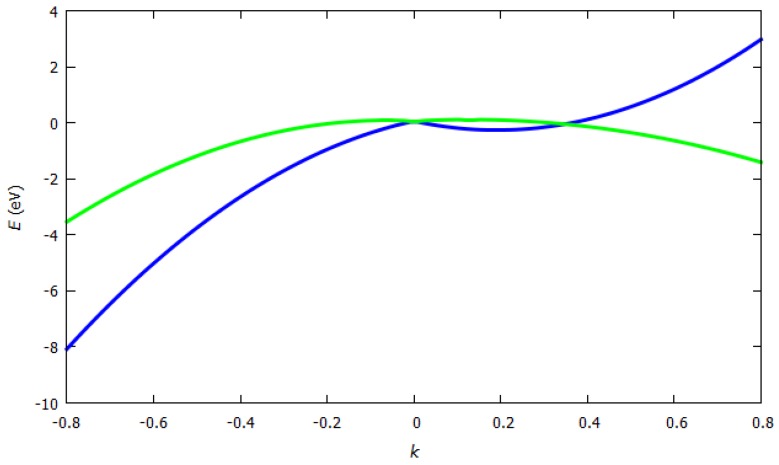
Tight binding dispersion relation for silicene.

**Figure 11 sensors-19-01295-f011:**
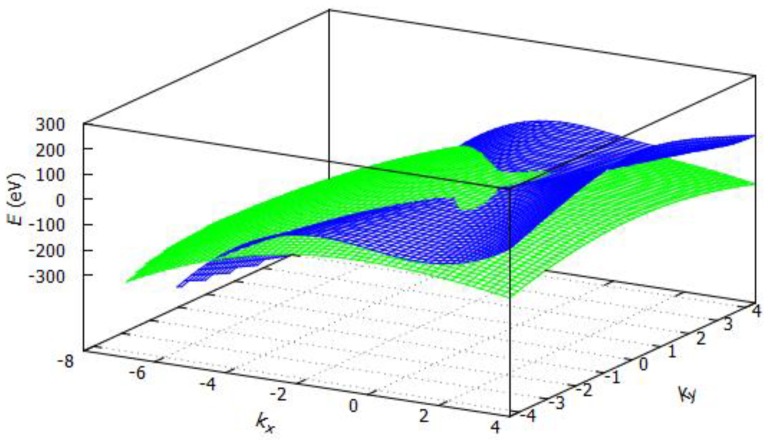
Silicene energy band structure.

**Figure 12 sensors-19-01295-f012:**
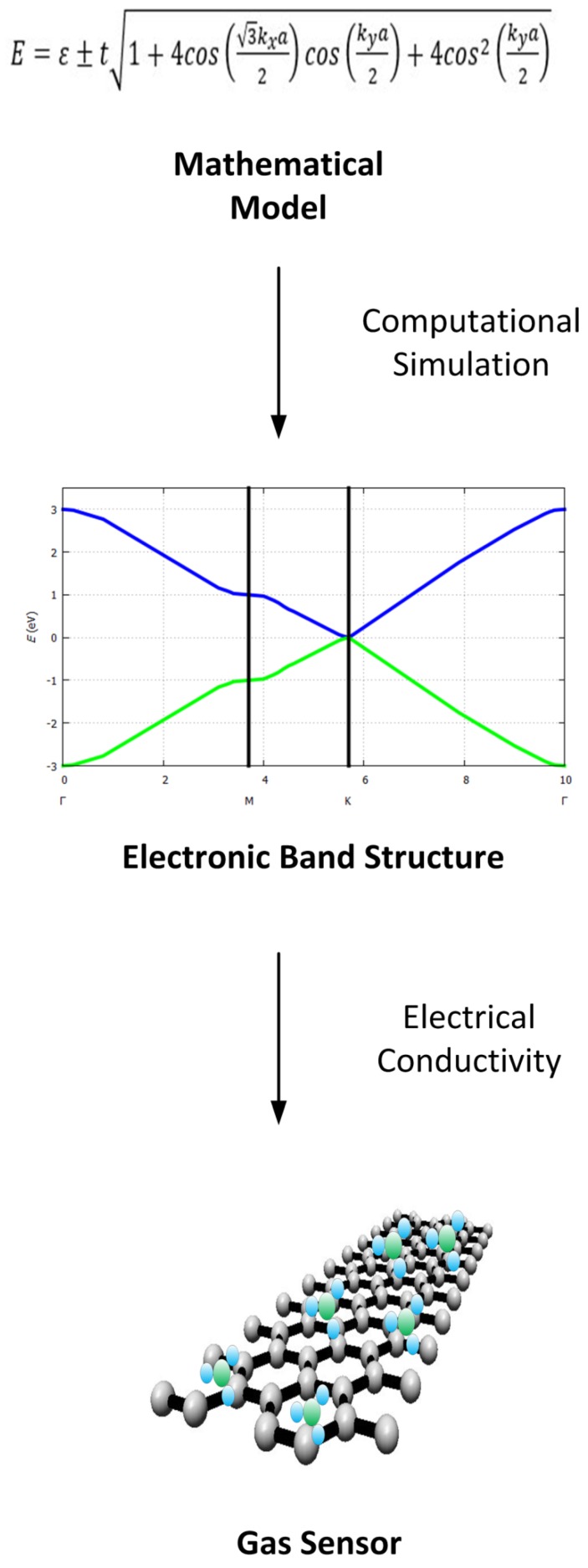
Methodology to design gas sensors based on two-dimensional materials.

**Figure 13 sensors-19-01295-f013:**
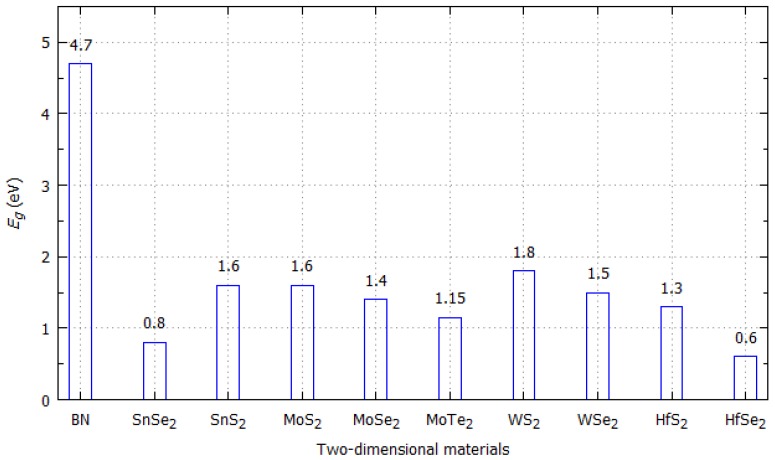
Band gaps of two-dimensional materials used in gas sensors.

**Figure 14 sensors-19-01295-f014:**
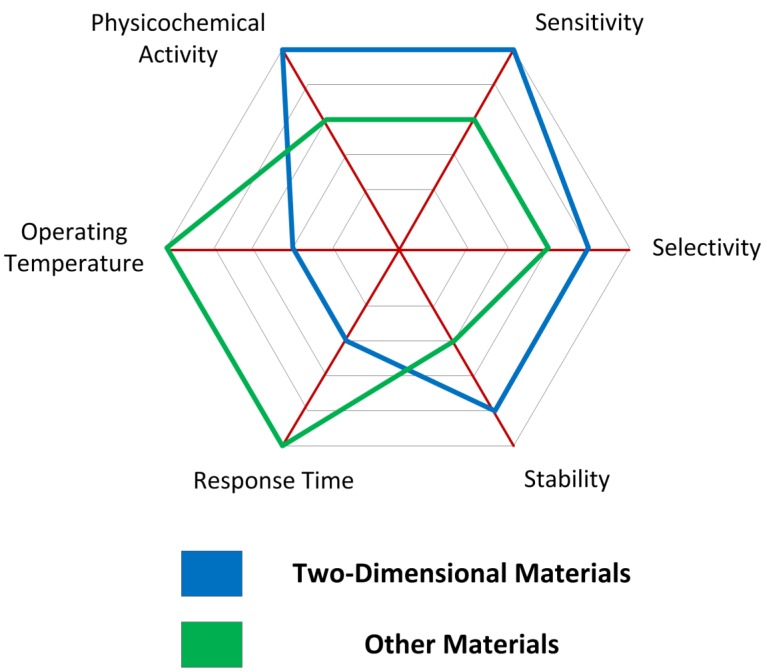
Key performance scores spider chart comparing the performance of gas sensors based on two-dimensional materials and those based on other materials.

**Table 1 sensors-19-01295-t001:** Two-dimensional materials based on one chemical element for electronic applications.

Two-Dimensional Material	BAND GAP	Electrical Properties	Crystal Structure	Unit Cell Parameters
Graphene	0 eV	Metal	Hexagonal	*a* = *b* = 0.2612 nm, *c* = 0.6079 nm, *α* = *β* = 90°, *γ* = 120°
Germanene	0.26 eV	Semimetal	Hexagonal	*a* = *b* = 0.249 nm, *c* = 0.268 nm, *α* = *β* = 90°, *γ* = 120°
Silicene	0.1 eV	Semimetal	Hexagonal	*a* = *b* = 0.382 nm, *c* = 0.45 nm, *α* = *β* = 90°, *γ* = 120°
Borophene (striped)	2 eV	Semimetal	Orthorhombic	*a* = 0.161, *b* = 0.286 nm, *c* = 0.911 nm, *α* = *β* = 90°, *γ* = 120°
Stanene	0.074 eV	Semimetal	Hexagonal	*a* = *b* = 0.468 nm, *c* = 0.283 nm, *α* = *β* = 90°, *γ* = 120°
Aluminene	1.618 eV	Semiconductor	Hexagonal	*a* = *b* = 0.449 nm, *c* = 0.259 nm, *α* = *β* = 90°, *γ* = 120°
Bismuthene	0.8 eV	Semimetal	Hexagonal	*a* = *b* = 0.449 nm, *c* = 0.259 nm, *α* = *β* = 90°, *γ* = 120°
Antimonene (β phase)	0.8–1.44 eV	Semimetal/Semiconductor	Hexagonal	*a* = *b* = 0.401 nm, *c* = 0.284 nm, *α* = *β* = 90°, *γ* = 120°

**Table 2 sensors-19-01295-t002:** Two-dimensional semiconductor materials for electronic applications (www.hqgraphene.com).

Two-Dimensional Material	Band Gap	Electrical Properties	Crystal Structure	Unit Cell Parameters
Diarsenic tritelluride As_2_Te_3_ (*α* phase)	0.2–0.3 eV	Semiconductor (indirect band gap), Topological insulator, Thermoelectric material	Monoclinic C	*a* = 1.430 nm, *b* = 0.403 nm, *c* = 0.986 nm, *α* = *γ* = 90°, *β* = 95.40°
Black phosphorus (BP)	0.3 eV	Semiconductor (direct band gap)	Orthorhombic C	*a* = 0.331 nm, *b* = 1.048 nm, *c* = 0.437 nm, *α* = *β* = *γ* = 90°
Hexagonal Boron Nitride (*h*-BN)	5.9 eV	Insulator/Semiconductor (direct band gap)	Hexagonal	*a* = *b* = 0.2502 nm, *c* = 0.6617 nm, *α* = *β* = 90°, *γ* = 120°
Dibismuth trisulphide (Bi_2_S_3_)	1.3–1.45 eV	Semiconductor (direct band gap)	Orthorhombic	*a* = 0.4025 nm, *b* = 1.117 nm, *c* = 1.135 nm, *α* = *β* = *γ* = 90°
Gallium sulfide GaS (*α* phase)	2.6 eV	Semiconductor (indirect band gap)	Hexagonal	*a* = 0.360 nm, *b* = 0.640 nm, *c* = 1.544 nm, *α* = *β* = 90°, *γ* = 120°
Gallium selenide GaSe (2H phase)	2.1 eV	Semiconductor (indirect band gap)	Hexagonal	*a* = *b* = 0.374 nm, *c* = 1.592 nm, *α* = *β* = 90°, *γ* = 120°
Germanium sulfide (GeS)	1.6 eV	Semiconductor (indirect band gap)	Orthorhombic	*a* = 1.450 nm, *b* = 0.364 nm, *c* = 0.430 nm, *α* = *β* = *γ* = 90°
Hafnium Disulfide (HfS_2_)	2 eV	Semiconductor (indirect band gap)	Hexagonal	*a* = *b* = 0.363 nm, *c* = 0.586 nm, *α* = *β* = 90°, *γ* = 120°
Hafnium Diselenide (HfSe_2_)	1.1 eV	Semiconductor (indirect band gap)	Hexagonal	*a* = *b* = 0.3745 nm, *c* = 0.616 nm, *α* = *β* = 90°, *γ* = 120°
Indium Selenide (In_2_Se_3_) (2H phase, *α*-phase)	1.14 eV	Semiconductor (direct band gap)	Hexagonal	*a* = *b* = 0.398 nm, *c* = 18.89 nm, *α* = *β* = 90°, *γ* = 120°
Molybdenum Disulfide (MoS_2_) (2H phase)	1.6 eV	Semiconductor (indirect band gap)	Hexagonal	*a* = *b* = 0.315 nm, *c* = 1.229 nm, *α* = *β* = 90°, *γ* = 120°
Molybdenum Ditelluride (2H phase)	1.2 eV	*n*-type Semiconductor (indirect band gap)	Hexagonal	*a* = *b* = 0.353 nm, *c* = 1.396 nm, *α* = *β* = 90°, *γ* = 120°
Molybdenum Diselenide (MoSe_2_) (2H phase)	1.2 eV	Semiconductor (indirect band gap)	Hexagonal	*a* = *b* = 0.329 nm, *c* = 1.289 nm, *α* = *β* = 90°, *γ* = 120°
Molybdenum Sulfide Selenide Alloy (MoSSe)	1.4 eV	Semiconductor (indirect band gap or direct band gap)	Hexagonal	*a* = *b* = 0.31–0.33 nm, *c* = 1.21–1.29 nm, *α* = *β* = 90°, *γ* = 120°
Molybdenum Tungsten Diselenide Alloy (MoWSe_2_)	1.2–1.3 eV	Semiconductor (indirect band gap)	Hexagonal	*a* = *b* = 0.31–0.33 nm, *c* = 1.21–1.30 nm, *α* = *β* = 90°, *γ* = 120°
Rhenium Disulphide (ReS_2_)	1.35 eV	Semiconductor (direct band gap)	Triclinic	*a* = 0.634 nm, *b* = 0.640 nm, *c* = 0.645 nm, *α* = 106.74°, *β* = 119.03°, *γ* = 89.97°
Rhenium Diselenide (ReSe_2_)	1.1 eV	Semiconductor (direct band gap)	Triclinic	*a* = 0.658 nm, *b* = 0.670 nm, *c* = 0.672 nm, *α* = 91.75°, *β* = 105°, *γ* = 118.9°
Antimony Telluride (Sb_2_Te_3_)	0.340–0.515 eV	Semiconductor (direct band gap), topological insulator, thermoelectric material	Hexagonal	*a* = *b* = 0.425 nm, *c* = 3.048 nm, *α* = *β* = 90°, *γ* = 120°
Tin Disulfide (SnS_2_) (2H phase)	2.2 eV	Semiconductor (indirect band gap)	Hexagonal	*a* = *b* = 0.364 nm, *c* = 0.589 nm, *α* = *β* = 90°, *γ* = 120°
Tin Diselenide (SnSe_2_)	2–3 eV	Semiconductor (indirect band gap)	Hexagonal	*a* = *b* = 0.381 nm, *c* = 0.614 nm, *α* = *β* = 90°, *γ* = 120°
Tantalum Disulfide (TaS_2_) (1T phase)	1 eV	Semiconductor (direct band gap), Charge density waves (CDW) system, Mott phase	Hexagonal	*a* = *b* = 0.336 nm, *c* = 0.590 nm, *α* = *β* = 90°, *γ* = 120°
Tungsten Disulfide (WS_2_) (2H phase	1.3 eV	Semiconductor (indirect band gap)	Hexagonal	*a* = *b* = 0.315 nm, *c* = 1.227 nm, *α* = *β* = 90°, *γ* = 120°
Tungsten Diselenide (WSe_2_)	1.3 eV	Semiconductor (indirect band gap)	Hexagonal	*a* = *b* = 0.328 nm, *c* = 1.298 nm, *α* = *β* = 90°, *γ* = 120°
Zirconium Diselenide (ZrSe_2_)	1 eV	Semiconductor (indirect band gap)	Hexagonal	*a* = *b* = 0.377 nm, *c* = 0.614 nm, *α* = *β* = 90°, *γ* = 120°
Zirconium Triselenide (ZrSe_3_)	1.1 eV	Semiconductor (indirect band gap)	Monoclinic P	*a* = 0.541 nm, *b* = 0.375 nm, *c* = 0.944 nm, *α* = *β* = 90°, *γ* = 97.50°

**Table 3 sensors-19-01295-t003:** Main two-dimensional materials used in gas sensing.

Two-Dimensional Material	Detected Gases	References
Graphene	CO, NO, NO_2_, NH_3_	[33]
	CO, NO	[34]
	NO_2_, NH_3_, H_2_, H_2_S, CO_2_, SO_2_	[35]
	NO_2_	[36]
Germanene	NH_3_, SO_2_, NO_2_	[37]
	N_2_, CO, CO_2_, NH_3_, NO, NO_2_, O_2_	[38]
	H_2_	[39]
	H_2_S, SO_2_, CO_2_	[40]
	NO_2_	[41]
	CO, NO	[42]
	N_2_, NO, NO_2_, NH_3_	[43]
Germanane	NH_3_	[44]
Silicene	NO, NO_2_	[45]
	NO	[46]
	H_2_S, SO_2_	[47]
Stanene	CO, NH_3_, H_2_S, O_2_, NO, NO_2_	[48]
	NO, NO_2_, NH_3_, N_2_O	[49]
	NH_3_, CO, NO, NO_2_	[50]
	NH_3_, NO_2_	[51]
Blue Phosphorene	O_2_, NO, SO_2_, NH_3_, NO_2_, CO_2_, H_2_S, CO, N_2_	[52]
Black Phosphorene	CH_3_OH	[53]
	NO_2_	[54]
	SO_2_	[55]
	CH_4_, CO_2_, H_2_, NH_3_	[56]
	PH_3_, AsH_3_	[57]
	HCN, HNC	[58]
	NO_2_	[59]
	SO_2_	[60]
Arsenene	NH_3_, NO_2_	[61]
	SO_2_, NO_2_	[62]
	NO, NO_2_	[63]
Aluminene	CO, NO	[64]
Antimonene	NH_3_, SO_2_, NO, NO_2_	[65]
	CO	[66]
	CO, NO, NO_2_, O_2_, NH_3_, H_2_	[67]
	NH_3_, NO_2_	[68]
Borophene	CO, NO, NO_2_, NH_3_, CO_2_	[69]
	NH_3_, NO, NO_2_, CO	[70]
WS_2_	NH_3_	[71]
	H_2_	[72]
	NH_3_, CH_2_O, CH_3_CH_2_OH, C_6_H_6_, C_3_H_6_O	[73]
WSe_2_	NO_2_, NH_3_, CO_2_, C_3_H_6_O	[74]
MoS_2_	NO	[75]
	NO_2_, NH_3_	[76]
	NO_2_	[77]
	CO, CO_2_, NO	[78]
	NH_3_, NO_2_	[79]
MoSe_2_	NO_2_	[80]
	CH_3_OH, CH_3_CH_2_OH	[81]
MoTe_2_	O_2_	[82]
Boron Nitride (BN)	CO	[83]
	CH_4_	[84]
	NH_3_	[85]
GeTe	NO	[86]
GeSe	NH_3_, SO_2_, NO_2_	[87]
	O_2_, NH_3_, SO_2_, H_2_, CO_2_, H_2_S, NO_2_, CH_4_, NO, CO	[88]
GeS	NO_2_	[89]
InN	CO, NH_3_, H_2_S, NO_2_, NO, SO_2_	[90]
InSe	CO, NH_3_, N_2_, NO_2_, NO, and O_2_	[91]
	CO, NO, NO_2_, H_2_S, N_2_, O_2_, NH_3_, H_2_	[92]
SnS_2_	NO_2_	[93]
	O_2_	[94]
	NH_3_	[95]
SnSe_2_	CH_4_	[96]
HfS_2_	O_2_	[82]
HfSe_2_	O_2_	[82]
M_2_CO_2_, M = Sc, Ti, Zr, and Hf	NH_3_	[97]
Ti_3_C_2_(OH)_2_	Volatile organic compounds (VOCs)	[98]
Sc_2_CO_2_	SO_2_	[99]
IrB_14_	CO, CO_2_	[100]

**Table 4 sensors-19-01295-t004:** Examples of hybrid or composites materials based on two-dimensional materials used in gas sensing.

Material	Detected Gases	References
Graphene/Molybdenum Disulfide (MoS_2_)	NO_2_	[101]
Indium Oxide (In_2_O_3_)—Graphene	NO_2_	[102]
Indium Oxide (In_2_O_3_)—Nitrogen-doped Reduced Graphene Oxide (N-RGO)	CO	[103]
Titania (TiO_2_)/Stanene	SO_x_	[104]
Palladium—Tin Oxide—Molybdenum Disulfide (Pd-SnO_2_/MoS_2_)	H_2_	[105]
Reduced Graphene Oxide—Zinc Oxide—Aluminum Gallium Nitride/Gallium Nitride (RGO-ZnO-AlGaN/GaN)	NO_2_, SO_2_, HCHO	[106]
Poly(3-hexylthiophene)—Zinc Oxide–Graphene Oxide (P3HT-ZnO@GO)	NO_2_	[107]

**Table 5 sensors-19-01295-t005:** Seven crystalline systems.

Crystal System	Relations	
	Lattice Constants	Interaxial Angles
Cubic	*a = b = c*	*α = β = γ =* 90°
Tetragonal	*a = b* ≠ *c*	*α = β = γ =* 90°
Orthorhombic	*a ≠ b ≠ c*	*α = β = γ =* 90°
Monoclinic	*a ≠ b ≠ c*	*α = γ*, *β ≠* 90°
Triclinic	*a ≠ b ≠ c*	*α ≠ β ≠ γ ≠* 90°
Trigonal (Rhombohedral)	*a = b = c*	*α = β = γ ≠* 90°
Hexagonal	*a = b ≠ c*	*α = β*, *γ =* 120°

**Table 6 sensors-19-01295-t006:** Critical points in the first Brillouin zone in 2D materials with hexagonal lattice.

Symbol	Description
Γ	Center of the Brillouin zone
A	Center of a hexagonal face
H	Corner point
K	Middle of an edge joining two rectangular faces
L	Middle of an edge joining a hexagonal and a rectangular face
M	Center of a rectangular face

**Table 7 sensors-19-01295-t007:** Crystalline lattices found in 2D materials.

Crystalline Lattice	First Brillouin Zone
Hexagonal	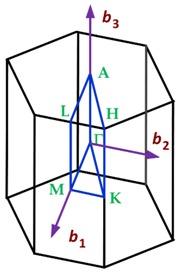
Orthorhombic	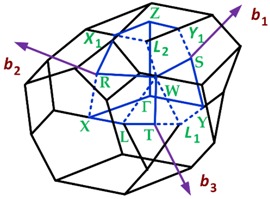
Monoclinic (P and C)	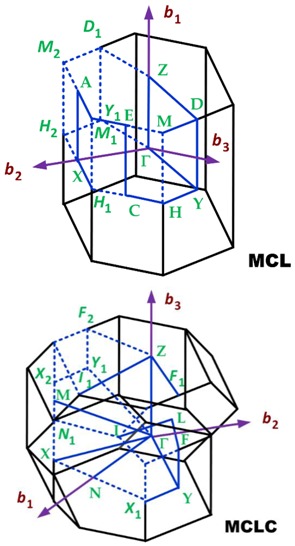
Triclinic	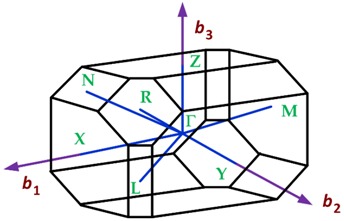

**Table 8 sensors-19-01295-t008:** Data of the first Brillouin zone of hexagonal lattices (http://lampx.tugraz.at/~hadley/ss1/bzones/).

Symmetry Points	[*k_x_*, *k_y_*, *k_z_*]	Point Group
Γ: (0, 0, 0)	[0, 0, 0]	6/*mmm*
A: (0, 0, 1/2)	[0, 0, π/*c*]	6/*mmm*
K: (2/3, 1/3, 0)	[4π/3*a*, 0, 0]	$\bar{6}$2*m*
H: (2/3, 1/3, 1/2)	[4π/3*a*, 0, π/*c*]	$\bar{6}$2*m*
M: (1/2, 0, 0)	[π/*a*, -π/$\sqrt{3}$*a*, 0]	*mmm*
L: (1/2, 0, 1/2)	[π/*a*, -π/$\sqrt{3}$*a*, π/*c*]	*mmm*

**Table 9 sensors-19-01295-t009:** Data of the first Brillouin zone of orthorhombic lattices (http://lampx.tugraz.at/~hadley/ss1/bzones/).

Symmetry Points	[*k_x_*, *k_y_*, *k_z_*]	Point Group
Γ: (0, 0, 0)	[0, 0, 0]	*mmm*
Y: (1/2, 1/2, 0)	[π/*a*, 0, 0]	*mmm*
Y’ or Y_1_: (−1/2, 1/2, 0)	[0, π/*b*, 0]	*mmm*
Z: (0, 0, 1/2)	[0, 0, π/*c*]	*mmm*
T: (1/2, 1/2, 1/2)	[π/*a*, 0, π/*c*]	*mmm*
T’ o T_1_: (−1/2, 1/2, 1/2)	[0, π/*b*, π/*c*]	*mmm*
S: (0, 1/2, 0)	[π/2*a*, π/2*b*, 0]	2/*m*
R: (0, 1/2, 1/2)	[π/2*a*, π/2*b*, π/*c*]	2/*m*

**Table 10 sensors-19-01295-t010:** Data of the first Brillouin zone of triclinic lattices.

Symmetry Points
Γ
L
M
N
R
X
Y
Z

**Table 11 sensors-19-01295-t011:** Data of the first Brillouin zone of P monoclinic lattices.

Symmetry Points	[*k_x_*, *k_y_*, *k_z_*]
Γ	[0, 0, 0]
X	[2π/*a*, −2π/*a*tan*γ*, 0]
Y	[0, 2π/*b*sin*γ*, 0]
Z	[0, 0, 2π/*c*]
A	[2π/*a*, −2π/*a*tan*γ*, 2π/*c*]
D	[0, 2π/*b*sin*γ*, 2π/*c*]

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
