# Peer review of "Electrical Properties of Two-Dimensional Materials Used in Gas Sensors"

_sensors, 2019, doi:10.3390/s19061295_

Round 1
Reviewer 1 Report
The paper presents a brief review on the electrical properties of two-dimensional materials used in gas sensors. The review provides the basic information regarding the concerned topic. I want to give major revisions to the author based on the comments provided in the next: 1- The paper discusses a detailed account of 2D materials' band gap engineering for gas sensors. However, as per the paper title, i recommend the authors to include the difference in the electrical properties of the 2D materials, such as change in resistance and responsivity/sensitivity, when exposed to detectable gases. 2- Section 2: It's better to support the explanation of some core concepts, such as what makes 2D materials advantageous to be used for gas sensors and the electronic and structural characteristics of 2D materials that make them the better candidate for gas sensing, with catchy pictorial representations. The concepts could be supported by adding schematic figures for readers' understanding. 3- Another hitch associated with the contemporary gas sensors is their room temperature sensing ability. I recommend to add a paragraph to glance at the room temperature gas sensing properties of 2D materials in the Review. 4- Provide bibliographic references to track the sources for all the equations used in the Review. 5- Section 4: Again, the explanation needs to be supported by alternative Figures for readers' understanding. Although the article provides a smooth flow of references, yet a good review paper requires the inclusion of all the components for its completeness. 6- Section 3 and Section 4 provide a detailed account of band gap engineering for 2D materials. However, the provided information needs to be linked with the gas sensor performance and results depending upon the specified band gap engineering, which is the prime objective of this review and is missing. Relate all the provided information in both sections with the gas sensor performance of 2D materials with proper referencing and Figures for readers' understanding and better flow.
Author Response
My response to comments of the reviewer 1 are added in a file.

Reviewer 2 Report
The author only arranges the discussion in the literature on the interaction of graphene and various compounds as a two-dimensional material from the viewpoint of electronic characteristics, and it is not interesting as content because there is no discussion about the following.
· Correlation between gas detection characteristics and band gap
· Correlation between gas detection and electronic structure
· Correlation between gas selectivity and electronic structure
Therefore, the present form is not suitable as a review article.
Author Response
My comments to the reviewer 2 are added in a file.

Reviewer 3 Report
The work is very complete as it gives a current overview on the use of 2D materials in the sensor field. In my opinion, it can be published in terms of content in the current form, after reviewing the text for possible spelling errors such as that reported in line 381.
Author Response
My comments to the reviewer 3 are added in a file.
